# 3D Numerical Study of External Axial Magnetic Field-Controlled High-Current GMAW Metal Transfer Behavior

**DOI:** 10.3390/ma13245792

**Published:** 2020-12-18

**Authors:** Lei Xiao, Ding Fan, Jiankang Huang, Shinichi Tashiro, Manabu Tanaka

**Affiliations:** 1School of Materials Science and Engineering, Lanzhou Jiaotong University, Lanzhou 730070, China; xxxiaolei@aliyun.com; 2State Key Laboratory of Advanced Processing and Recycling of Non-ferrous Metals, Lanzhou University of Technology, Lanzhou 730050, China; 3School of Materials Science and Engineering, Lanzhou University of Technology, Lanzhou 730050, China; 4Joining and Welding Research Institute, Osaka University, Osaka 5670047, Japan; tanaka@jwri.osaka-u.ac.jp

**Keywords:** gas metal arc welding, high-efficiency, external axial magnetic field, numerical simulation, metal rotating-spray transfer, spatter

## Abstract

For gas metal arc welding (GMAW), increasing the welding current is the most effective way to improve welding efficiency. However, much higher current decreases the welding quality as a result of metal rotating-spray transfer phenomena in the high-current GMAW process. In this work, the external axial magnetic field (EAMF) was applied to the high-current GMAW process to control the metal transfer and decrease the welding spatters. A unified arc-droplet coupled model for high-current GMAW using EAMFs was built to investigate the metal rotating-spray transfer behavior. The temperature fields, flow fields in the arc, and droplet were revealed. Considering all the heat transferred to the molten metal, the Joule heat was found to be the dominant factor affecting the droplet temperature rise, followed by the anode heat. The conductive heat from the arc contributed less than half the value of the other two. Considering the EAMFs of different alternating frequencies, the arc constricting effects and controlled metal transfer behaviors are discussed. The calculated results agree well with the experimental high-speed camera observations.

## 1. Introduction

Gas metal arc welding (GMAW) is a high-efficiency welding method. There are many different metal droplet transfer modes with various welding currents for wire GMAW of mild steel, which is important for the weld quality [1]. Axial projected/streaming spray transfer has been widely used and studied for its stability and high quality. However, the metal transfer process changes to rotating-spray transfer once the welding current changes from the first critical current to the second critical current. It has been reported that the metal rotating-spray transfer process can lower the weld quality due to its heavy spatter. For this reason, it is avoided in industrial applications and there are only a few related studies in the literature [2,3,4]. However, the welding efficiency is greatly improved if the metal rotating-spray transfer can be controlled by constricting the welding arc and decreasing the deflected angle of the rotating molten wire from the solid wire axis (hereinafter, defined as a rotating deflected angle) in high-current single-wire GMAW. The Transferred Ionized Molten Energy (TIME) welding method uses mixed shielding gases of four different gases (oxygen, carbon dioxide, argon, helium) of different proportions to constrict the welding arc and metal rotating deflected angle of high-current GMAW [5]. However, the TIME welding method is limited in its industrial applications due to the strict requirement of four matching shielding gases. Nakamura et al. developed a welding process that uses a small amount of oxygen and a coaxial multilayer solid wire to reduce the length of the metal liquid column [4,6]. However, its welding torch structure is complex and difficult to operate.

The magnetically controlled method, which is of low cost and easy to operate, has been applied in arc welding for decades, because there are current flows in the welding arc, electrode, droplet (the droplet includes the molten wire tip and the free droplet), and weld pool, which can be influenced from the external magnetic field (EMF) to generate external electromagnetic forces [7,8,9,10]. Wu et al. summarized the recent studies on the influence of EMF on the arc welding process in a review article [11]. The behaviors of arc and molten metal can be controlled flexibly by designing the spatiotemporal distribution of the EMF. Nomura et al. applied a cusp-type EMF to tungsten inert gas (TIG) welding to change the cross-section of the arc from a circular to an elliptical shape, achieving high-speed welding without defects [12]. Chen et al. studied the TIG arc with an external axial magnetic field (EAMF) and found that the EAMF induced the arc rotation and resulted in a magnetohydrodynamic (MHD) pumping effect, which shrank the arc attachment [13]. Chang et al. used an external longitudinal magnetic field during short-circuit GMAW, improving the metal transfer frequency, and decreasing spatter generation [14]. Wang et al. prevented the occurrence of a humping bead in high-speed GMAW by applying an external transverse magnetic field [15]. Chen et al. successfully suppressed undercut defects in high-speed GMAW using an assisting compound magnetic field [16].

In this work, we used an alternating EAMF to control the high-current GMAW metal rotating-spray transfer process with the help of the external forces in the arc and droplet. For this study of GMAW metal transfer behavior with an EAMF, the acting forces: heat and mass transfer in the arc and droplet, are the key points, which are very difficult to investigate using only experimental approaches. Thus, a numerical simulation using high-speed video observation for verification is one of the most effective research methods for the GMAW metal transfer process. Ogino et al. provided a 3D acting force model of the fluid molten wire in the high-current GMAW metal transfer process [2]. It was concluded that the concentration of the magnetic field lines made the electromagnetic force higher on one side than the other due to a bending molten liquid column, and for this reason, the molten liquid column continued to sway. However, this proposed numerical model was not coupled with the arc, so the interaction behavior between the arc and the droplet is still not clear. In recent years, most arc-droplet coupled models have demonstrated 2D issues in terms of GMAW globular transfer and axial spray transfer. Cadiou et al. provided an arc-droplet coupled GMAW metal transfer numerical model for learning wire arc additive manufacturing (WAAM), in which the material supply/cooling cycles between each layer were simulated and experimentally measured [17]. Wang et al. proposed a comprehensive numerical model containing the arc and filler metal to investigate the impacts of the external compound magnetic field on the arc and droplet behavior [7]. However, the metal vapor behavior, which is very important for the formation of the electric conduction path, was not considered in their mathematical models.

In the present work, a 3D unified arc-droplet coupled model for high-current GMAW, which takes into account the influence of metal vapor and alternating EAMFs, was constructed for the first time in order to investigate the metal rotating-spray transfer behavior under the application of alternating EAMFs. The external electromagnetic force and large changes in temperature and flow fields in both the arc plasma and molten metal are discussed in detail. This is a necessary foundation for further studies and applications of magnetic-field-controlled GMAW technology.

## 2. Mathematical Model

The details of the 3D mathematical model concerning the assumptions and governing equations in this work are similar to those of the 2D model in our previous work [18]. To track the free surface, a multifluid volume of fluid (VOF) model was used. Two independent governing equations of mass, momentum, and energy for the gas phase (arc plasma) and metal phase (fluid metal), respectively, were solved to acquire the heat and mass transfer process of the GMAW in the rotating-spray transfer mode. Furthermore, the main and auxiliary equations also include the current continuity equation, a set of Maxwell equations, the VOF equation, and the metal vapor transport equation (only iron vapor is considered). The main equations are as follows:

Fluid volume fraction:(1)$$\frac{\partial F_{m}}{\partial t}+\nabla \cdot (F_{m}{\vec{v}}_{m})=0$$
where *F*_g_ = 1 − *F*_m_.

Mass continuity:(2)$$\frac{\partial \rho_{i}}{\partial t}+\nabla \cdot (\rho_{i}{\vec{v}}_{i})=0$$

Momentum conservation:(3)$$\frac{\partial (\rho_{i}\;{\vec{v}}_{i})}{\partial t}+\nabla \cdot (\rho_{i}{\vec{v}}_{i}{\vec{v}}_{i})=-\nabla P+\nabla \cdot \vec{\vec{\tau }}+\vec{j}\times \vec{B}+{\vec{S}}_{u}$$

Energy conservation:(4)$$\frac{\partial (\rho_{i}\;h_{i}F_{i})}{\partial t}+\nabla \cdot (\rho_{i}{\vec{v}}_{i}h_{i}F_{i})=\nabla \cdot (k_{i}\nabla T_{i})+\frac{j^{2}}{\sigma_{i}}F_{i}+S_{T}$$

Metal vapor mass conservation:(5)$$\frac{\partial (\rho_{g}CF_{g})}{\partial t}+\nabla \cdot (\rho_{g}C{\vec{v}}_{g}F_{g})=\nabla \cdot (D\nabla CF_{g})+M_{\mathrm{vap}}$$
where *F* is the volume fraction; subscript i indicates a phase (*g*: gas phase, *m*: metal phase); *t* is the time; $\vec{v}$ is the velocity vector; *ρ* is the density; *P* is the static pressure; $\vec{\vec{\tau }}$ is the viscous shear tensor; $\vec{j}$ is the current density vector; *B* is the magnetic flux density vector calculated from the vector potential *A*, including a self-induced and externally applied one; *h* is the specific enthalpy; *k* is the thermal conductivity; *T* is the temperature; *σ* is the electric conductivity; *C* is the mass fraction concentration of metal vapor; and *D* is the diffusion coefficient of iron metal vapor in argon shielding gas [19]. *M*_vap_ is expressed as the mass source of the metal vapor and was calculated by using the Hertz–Knudsen–Langmuir equation [20].

The source terms in the momentum conservation equation are different for each phase.

In the gas phase:
(6)$${\vec{S}}_{u}=0$$

In the metal phase:(7)$${\vec{S}}_{u}=\rho_{m}\vec{g}+\mu_{g}\frac{\partial {\vec{v}}_{g}}{\partial s}\times \left|\nabla F_{m}\right|+\gamma k_{\mathrm{cur}}\nabla F_{m}$$
where *μ*_g_ is the gas phase dynamic viscosity; ***s*** is the tangential normal vector to the free surface; *γ* is the surface tension coefficient; *k*_cur_ is the curvature. The three terms are in respect to gravity, arc plasma shear stress, and surface tension force.

The source terms in the energy conservation equation including evaporating heat and the heat exchange between the phases are expressed, respectively.

In the gas phase:(8)$$S_{T}=M_{\mathrm{vap}}c_{p}T_{m}-\int_{T_{m}}^{T_{g}}k_{g}dT_{g}/\delta_{\mathrm{gm}}\left|\nabla F_{m}\right|$$

In the metal phase:(9)$$S_{T}=-M_{\mathrm{vap}}(c_{p}T_{m}+h_{\mathrm{vap}})+\int_{T_{m}}^{T_{g}}k_{g}dT_{g}/\delta_{\mathrm{gm}}\left|\nabla F_{m}\right|+\left|\frac{\vec{j}}{e}\cdot \nabla F_{m}\right|\phi_{a}$$
where *c*_p_ is the gas phase-specific heat; *δ*_gm_ is the thickness of the mixture region; *h*_vap_ is the vaporization heat; *e* is the elementary charge; *ϕ*_a_ is the work function of the anode material.

Here, the calculation domain and boundary conditions are described. The 3D calculation domain was designed as a column with a diameter of 24 mm as shown in Figure 1. The length and diameter of solid mild steel wire were 2 mm and 1.2 mm, respectively. The arc length between the solid wire tip and the base metal surface was 10 mm. As assumed in [18], the solid wire region, whose temperature was lower than the melting point of wire material, was treated as a fluid phase at high viscosity, hence there are two different fluid phases in the calculation domain: arc plasma and fluid metal.

On the top boundaries of the domain, the metal inlet and gas inlet are defined, giving the wire feed rate *v*_min_ and shielding gas flow rate *v*_gin_. The direct current (DC) of 400 A was also applied to the metal inlet. The side boundary of the domain was the gas pressure outlet boundary, giving the atmospheric pressure *P*_ATM_; the bottom boundary was the ground wall where the electric potential *Φ* was 0 and the temperature was 3000 K. At that temperature, the arc plasma electric conductivity is not 0, hence the current can be continuous through the cathode boundary based on the simplification of the ‘Local thermodynamic equilibrium diffusion approximation’ [21]. This bottom boundary temperature was assumed to maintain the valid electrical conductivity of the gas in contact with the bottom boundary. In addition, the sidewall of the solid wire with a length of 2 mm from the top was defined to be a coupled boundary for the energy conservation equations and Maxwell equations. The boundary conditions are shown in detail in Table 1.

The electromagnetic coil excited by a self-made excitation source was covered on the coaxial welding torch nozzle to generate the EAMF [18]. The EAMF was assumed to have a uniform spatial distribution in the entire calculation domain. The frequency of the EAMF was set to 0 Hz (constant magnetic field), 100 Hz, 200 Hz, or 500 Hz, and its strength was 0.02 T. The magnetic flux density waveform of the alternating EAMF was simplified to be a square wave. Taking 100 Hz as an example, the axial magnetic flux density waveform of 100 Hz is shown in Figure 2.

The main acting forces of gravity, surface tension, arc plasma shear stress, electromagnetic force, and arc plasma pressure were considered in the molten metal phase, while only the electromagnetic force was applied in the gas phase of arc plasma. For the metal vapor including only iron vapor in the arc plasma, the diffusion coefficient based on the second viscosity approximation method was applied according to Murphy [19]. For the meshing of the whole calculation domain, only hexahedral structured meshes were used to ensure an accurate simulation, and the meshes near the metal liquid column boundary were refined. There are a total of 102,240 cells in the calculated domain of 5367 mm^3^. The thermophysical transport properties and net emission coefficients of the Ar-Fe mixture plasma are referenced from [22,23]. The welding parameters are listed in Table 2.

## 3. Results and Discussions

### 3.1. High-Current GMAW Metal Transfer Behavior without EAMFs

Firstly, high-current GMAW metal transfer without EAMFs is discussed. The GMAW metal transfer shifted to rotating-spray transfer with a diameter of 1.2 mm solid mild steel wire and pure argon shielding gas when the welding current reached about 400 A. The metal rotating direction was clockwise along the *z*-axis.

The arc temperature field and metal rotating pattern in one cycle is shown in Figure 3. The maximal arc temperature was about 14,000 K, and the maximal temperature region was distributed near the side surface of the upper molten metal. The arc plasma rotated together with the helically rotating metal liquid column. In the work of Nomura et al., the arc temperature of spray transfer (not rotating-spray transfer) GMAW using the pulsed current with a peak current of 440 A was measured with the spectroscopic method and the maximal arc temperature, which appeared during the peak current in the argon arc region, was around 15,000 K [24]. In the case of the rotating spray transfer, the low-temperature gas in the traveling direction of the arc has to be heated continuously with the rotation to produce a new current path, which is considered to have a cooling effect on the arc, hence there is little difference between the two cases.

The droplet temperature field is shown in Figure 4. The maximal droplet temperature was about 2200 K, ignoring the droplets in contact with the weld pool, which was set to 3000 K. Mamat et al. measured the droplet temperature in conventional metal inert gas welding (MIGW) with the two-color temperature measurement method and concluded that the droplet maximal temperature was about 2400 K at 160 A in globular MIGW and 2700 K at 230 A in streaming spray MIGW [25]. The reason the maximal droplet temperature decreased to 2200 K for 400 A rotating spray transfer in our results is probably that the rotating motion dispersed the arc plasma and promoted metal transfer, decreasing the droplet heating time. The heat input of the droplet came from a Joule heat of about 688 W, a conductive heat of about 208 W, and an anode heat of about 523 W. The maximal heat input was the Joule heat since the current density in the molten metal was large and the liquid column was long.

The metal vapor evaporated from the droplet surface, cooling the arc due to strong radiation loss. Therefore, the arc temperature below the droplet became lower. The maximal mass fraction of metal vapor was about 90%, and its distribution is shown in Figure 5.

Figure 6 demonstrates the arc plasma flow velocity vector. The maximal velocity was about 200 m/s beside the metal liquid column. This kind of arc plasma flow pattern drove the heat and metal vapor downwards and outwards.

The arc pressure field is shown in Figure 7. The arc static pressure on the free droplet upper surface reached about 800 Pa since the arc plasma flow hit against it. In the outside region, the arc static pressure decreased since the arc plasma flow velocity was weakened.

It can be seen in Figure 8 that the maximal metal flow velocity was about 5 m/s and the average velocity was around 2.5 m/s. The latter is almost in agreement with the velocity found in the droplet during pulsed MIGW with a peak current of 420 A [20]. The smaller the free droplet is, the faster the transfer, and thus the spatters usually have a higher traveling speed.

### 3.2. High-Current GMAW Metal Transfer Behavior with EAMFs

To control the rotating deflected angles as well as the sizes and amounts of spatters in high-current GMAW, we used EAMFs at different alternating frequencies to provide alternating angular forces on the arc and molten metal to improve the metal transfer process. In Figure 9**,** we can see that there were four different EAMFs at frequencies of 0 Hz (i.e., this denotes a constant EAMF), 100 Hz, 200 Hz, and 500 Hz in this work. As is shown in Figure 9a, the metal liquid column rotated in a clockwise direction, and the free droplet transferred off the axis together with small amounts of spatter formed at the metal liquid column tips. The rotating angle increased and the free droplet size decreased as a 0 Hz EAMF was applied. The rotating frequency was about 350 Hz, which is almost the same as without the magnetic field. The 100 Hz EAMF caused the metal liquid column to lose its rotating direction. The metal liquid column changed its rotating direction at a frequency of 100 Hz, and the rotating angle decreased to a large degree at that moment of change. As the magnetic field alternating frequency increased to 200 Hz, the metal liquid column did not change the rotating direction because of the large inertia of the molten steel. In the previous half-cycle, the metal liquid column rotating motion was intensified and was restrained in the left half-cycle, i.e., vice versa. As shown in Figure 9e, when the magnetic field alternating frequency was much larger, the metal liquid column rotated at a stable angle. The low-frequency magnetic fields were found to be more suitable in terms of controlling the metal liquid column by decreasing the rotating deflected angle and welding spatters.

The change in the metal liquid column rotating frequency and angle caused by the EAMF is mainly due to changes in the metal flow pattern caused by the external electromagnetic force. Thus, the movement state of the whole metal liquid column changes. The EAMFs not only affect the flow behavior of the metal liquid column but also affect the arc behavior. As shown in Figure 10, the maximal arc temperature and flow velocity in high-current GMAW fluctuated within a certain range in one cycle due to the instability of the arc. This fluctuation is expressed by the standard deviations, and it was found that the highest fluctuations in arc temperature and flow velocity when the low-frequency 100 Hz magnetic field was applied. When the external magnetic field frequency was low, the rotating direction of the metal liquid column periodically changed. Hence, the droplet could transfer at a small angle at the moment of change. As shown at 3 ms and 8 ms in Figure 9c, the arc deflected angle became smaller as shown in Figure 10, the arc temperature and flow velocity increased. Regardless of the EAMF frequency, the arc temperature and the arc plasma flow velocity became higher than without the EAMF.

The welding arc is quite different from the metal, especially in terms of the physical properties. The arc plasma density is less than 1/70,000 of the metal (the Ar plasma temperature is assumed to be 15,000 K); therefore, the arc plasma has very small inertia, which leads to the arc instantly changing rotating direction even if the additional EAMF frequency increases to thousands of Hz. Meanwhile, its angular velocity can still periodically change. In previous studies [26,27,28], the application of EAMF was found to cause a rotating movement in the arc. This is because the lower internal pressure than the external pressure due to the centrifugal force causes the arc to constrict. This causes the arc current path to becoming fixed toward the central axis, which also affects the current path in the metal liquid column due to the current continuity. Hence, the constricted arc is thought to further restrict the rotating motion of the metal liquid column to improve the EAMF-controlled effect.

Figure 11 shows the high-current GMAW arc velocity vector at selected moments with EAMFs being applied. When there was no EAMF, as shown in Figure 11a, the arc plasma began to accelerate at the welding wire tip. Combined with the flow field on the x-y plane, it was found that the arc velocity had almost no angular component. The arc plasma flowed directly to the surface of the base metal and reached the maximal flow velocity of 200 m/s around the lower end of the metal liquid column. When the 0 Hz EAMF was applied, the maximal arc speed reached 270 m/s. Meanwhile, the arc constricted significantly when it rotated at high speed under the metal liquid column. After applying the alternating EAMFs, the arc rotating direction also changed periodically due to the periodic reversal of the angular electromagnetic force; the arc significantly constricted when the rotating deflected angle of the metal liquid column was reduced, and then flowed to the tip of the metal liquid column and ultimately to the base metal. It is obvious that the arc plasma velocity decreases when the frequency of the applied magnetic field increases since the inertia effect.

On the one hand, the 0 Hz EAMF caused the arc to constrict and rotate. On the other hand, when there was a radial component of current in the metal liquid column, the EAMF produced an angular electromagnetic force to rotate it. The rotating metal liquid column was lifted due to the centrifugal force, which depended on the angular velocity. As a result, the rotating deflected angle increased. However, an alternating EAMF prevented an increase in the angular velocity of the rotating molten wire, so the rotating deflected angle decreased.

It can be concluded that the higher frequency magnetic field controls the high-current GMAW metal transfer through the magnetic field-controlled arc’s constricting effect. In addition to this effect, the low-frequency magnetic field can also inhibit the metal rotating-spray transfer by periodically changing the rotating direction or rotating intensity of the metal liquid column.

### 3.3. Experimental Verification

To verify the calculated results, for the metal transfer process, in particular, high-speed photography technology was used. In the comparative experimental process, the welding current was adjusted to 400 A by setting the welding voltage at 40 V, the wire feed rate at 20 m/min, and the wire stick-out length at 20 mm. The argon shielding gas flow rate was 20 L/min, and the welding speed was 0.5 m/min. The exciting current in the magnet coil (with 120 turns) was set to 10 A to acquire a similar magnetic flux density to the numerical model in the arc region.

The influence of alternating magnetic field frequency on high-current GMAW metal transfer behavior is shown in Figure 12. The rotating frequency, as shown in Figure 12a, without EAMF was nearly 500 Hz, which is somehow higher than the calculated results. In the mathematic model, the weld pool acting as a cathode is not considered. In the case without EAMF, the rotation of the molten wire was thought to be primarily induced by the arc disturbance [11], so the random movements of cathode spots, which were ignored in this model, were predicted to affect the rotating frequency. After applying the 0 Hz EAMF, the rotating deflected angle increased, the arc was compressed along the axial direction, and the arc length was reduced to half the value without EAMF. The entire bright arc region no longer rotated together with the metal liquid column but stayed close to the axis. The rotating frequency of the metal liquid column remained unchanged at 500 Hz. When the low-frequency 100 Hz EAMF was applied, the rotating direction of the metal liquid column was no longer in a single direction but rotated back and forth periodically, and the rotating deflected angle decreased significantly in a short time when the rotating direction changed. This is in good agreement with the previously calculated results. When the frequency of the EAMF was 200 Hz, the rotating direction of the metal liquid column no longer alternated between the clockwise and anti-clockwise directions. The EAMF promotes a rotating motion in the first half cycle and inhibits a rotating motion in the second half cycle. When the 500 Hz EAMF was applied to the whole welding arc region, it was found that there were no positive or negative rotating phenomena, neither were there rotating strengthening and suppressing phenomena.

## 4. Conclusions

A previously created unified arc–droplet coupled model was developed for 3D high-current GMAW to investigate the metal rotating-spray transfer behavior. This novel high-efficiency GMAW method utilizes the effect of EAMFs, and its heat and momentum transfer processes are discussed in detail. The main conclusions of this work are as follows:(1)Regarding the three main heat inputs to the droplet during the metal rotating-spray transfer process in high-current GMAW with pure argon shielding gas, the Joule heat is the dominant factor affecting the droplet temperature rise, followed by the anode heat. The conductive heat from the arc contributed less than half the value of the other two.(2)The EAMFs can enhance the maximal arc temperature and maximal flow velocity during high-current GMAW regardless of the EAMF frequency.(3)A constant EAMF causes the high-current GMAW metal liquid column to rotate in a fixed direction and increases the rotating deflected angle. A 100 Hz EAMF changes the rotating direction of the metal liquid column synchronously and decreases the rotating deflected angle at the moment of change. The rotating direction of the metal liquid column does not change in the case of 200 Hz, but the rotating motion in the first half cycle is suppressed, and the rotating motion in the second half cycle is strengthened. A 500 Hz EAMF does not affect the rotating direction of the metal liquid column neither does it strengthen or suppress the rotating cycles.

## Figures and Tables

**Figure 1 materials-13-05792-f001:**
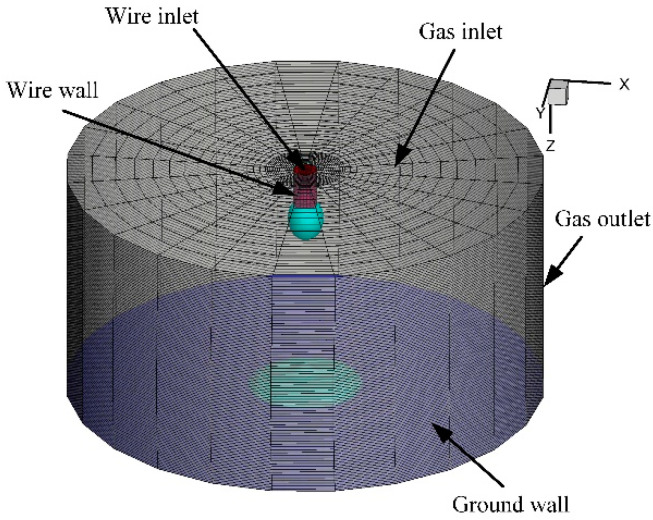
Schematic diagram of the 3D calculation domain.

**Figure 2 materials-13-05792-f002:**
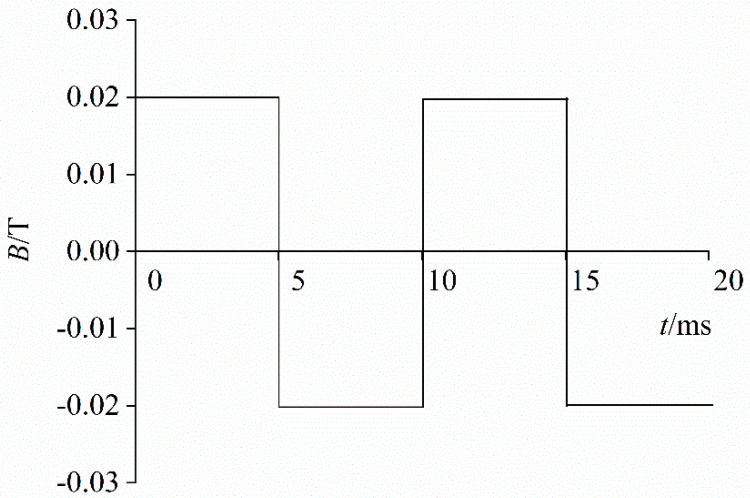
Magnetic flux density waveform of 100 Hz external axial magnetic field (EAMF).

**Figure 3 materials-13-05792-f003:**
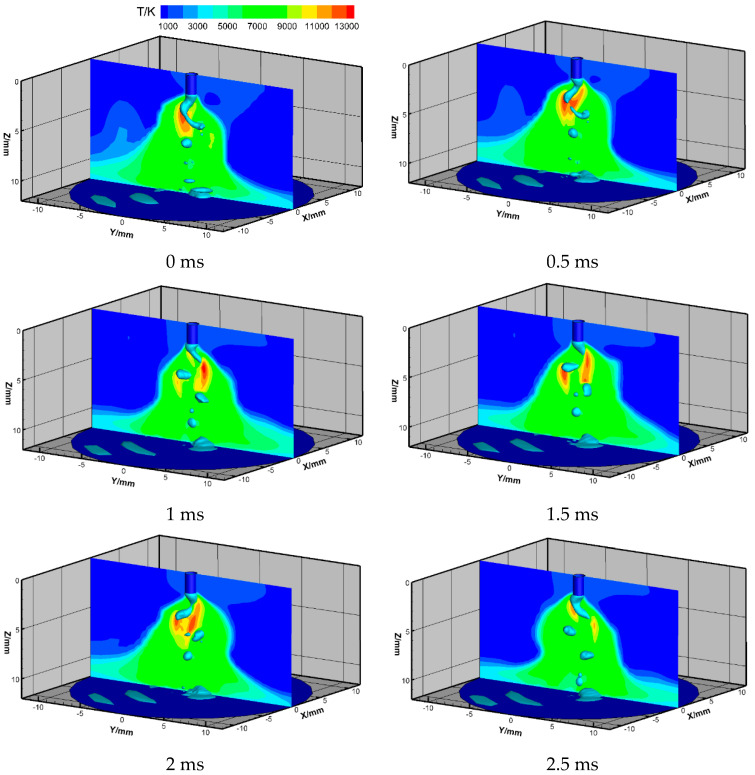
Arc temperature field for high-current gas metal arc welding (GMAW).

**Figure 4 materials-13-05792-f004:**
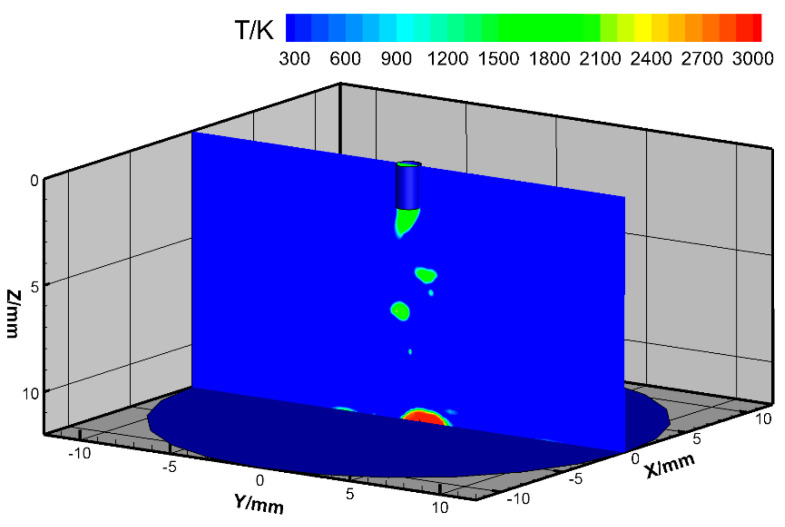
Droplet temperature field for high-current GMAW.

**Figure 5 materials-13-05792-f005:**
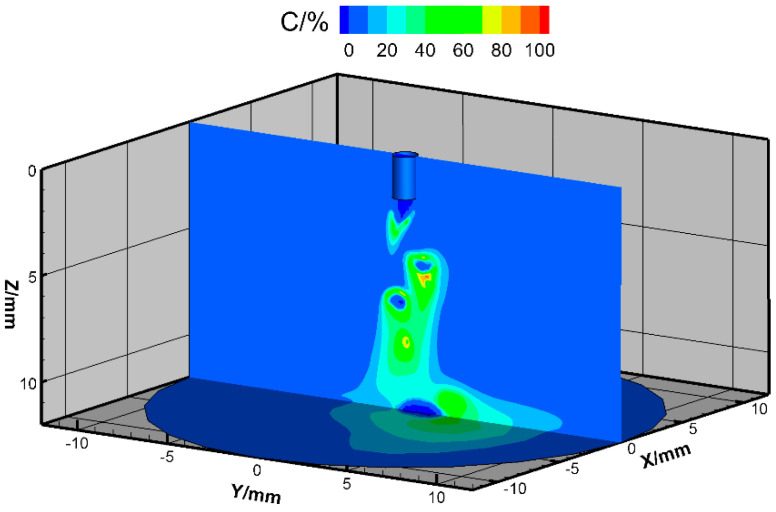
Metal vapor distribution for high-current GMAW.

**Figure 6 materials-13-05792-f006:**
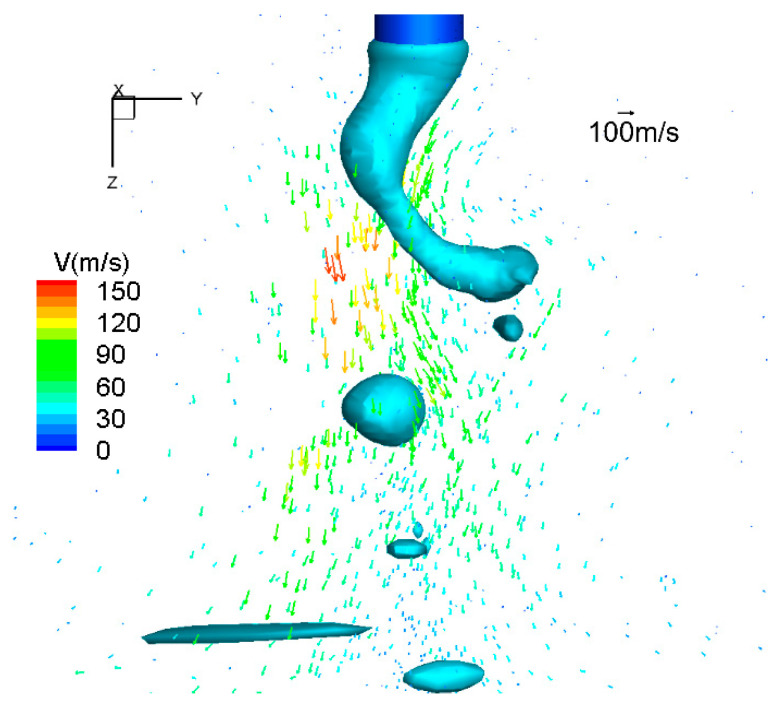
Arc plasma flow velocity vector for high-current GMAW.

**Figure 7 materials-13-05792-f007:**
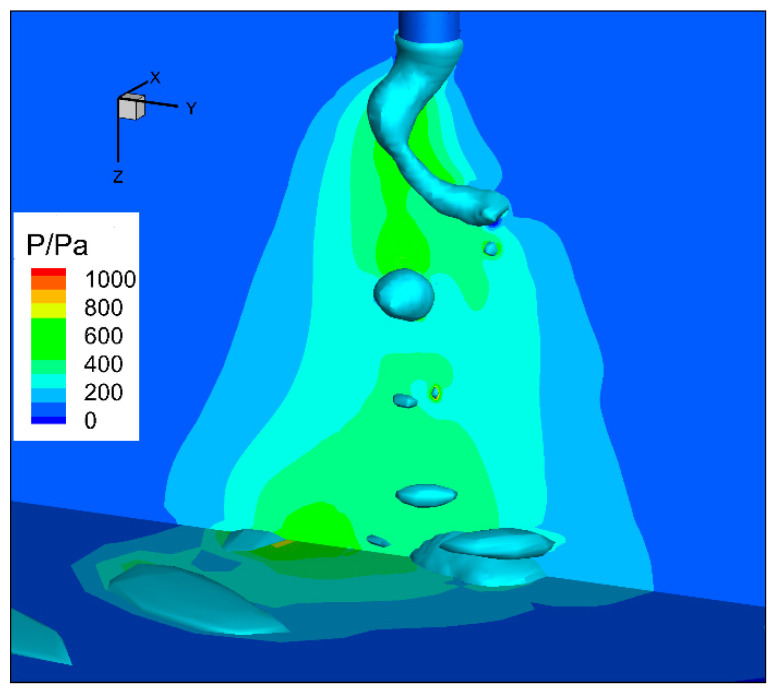
Arc pressure field for high-current GMAW.

**Figure 8 materials-13-05792-f008:**
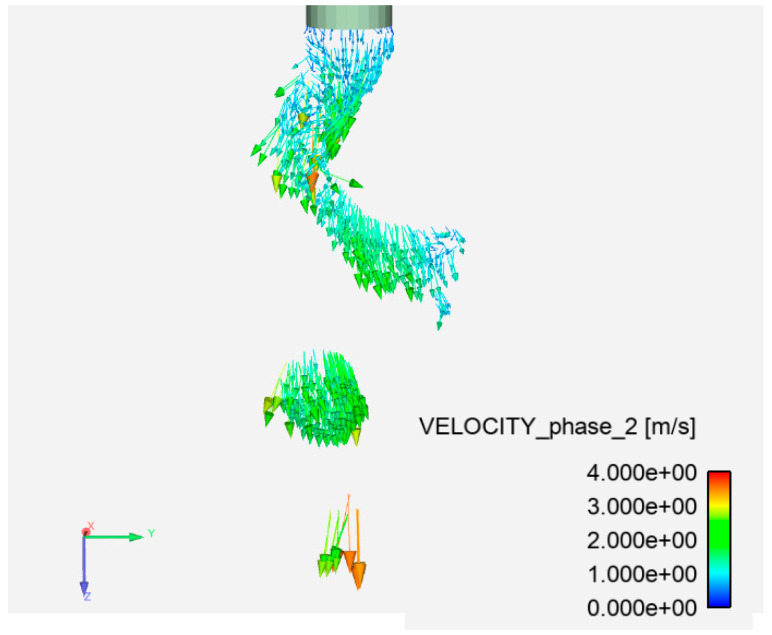
Droplet metal velocity vector for high-current GMAW.

**Figure 9 materials-13-05792-f009:**
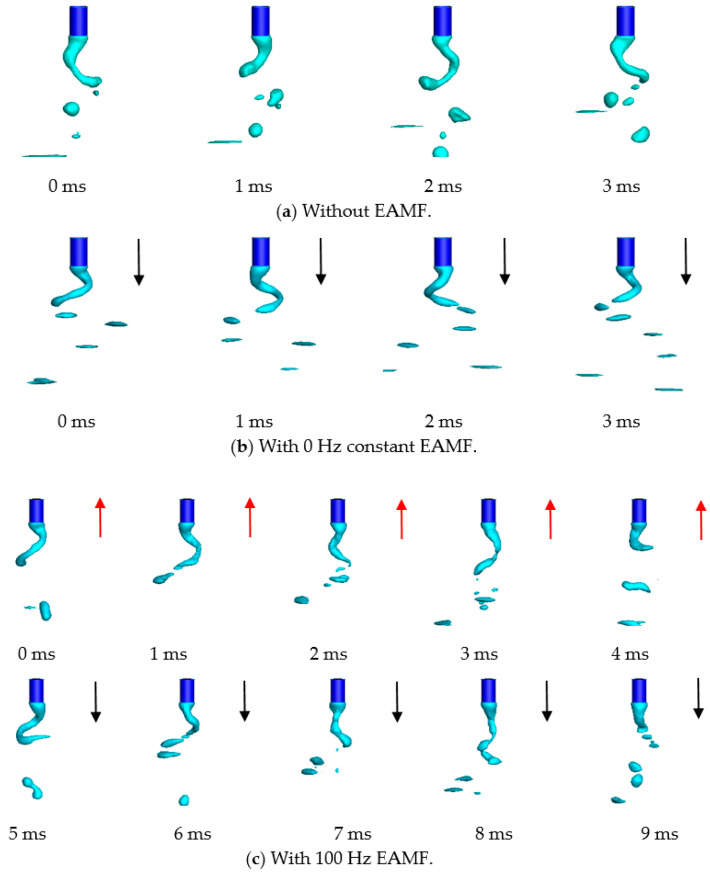
High-current GMAW metal transfer under the influence of EAMFs at different alternating frequencies.

**Figure 10 materials-13-05792-f010:**
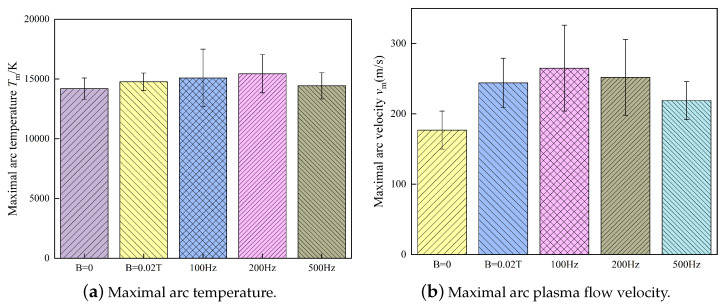
Maximal arc temperature and flow velocity in high-current GMAW with EAMFs at different alternating frequencies.

**Figure 11 materials-13-05792-f011:**
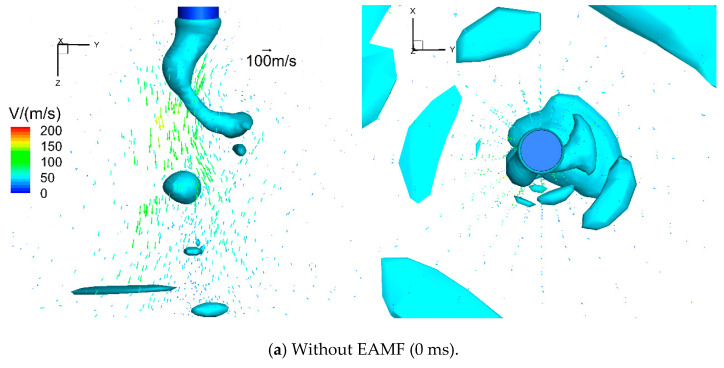
The arc plasma flow velocity vector for high-current GMAW under the influence of EAMF.

**Figure 12 materials-13-05792-f012:**
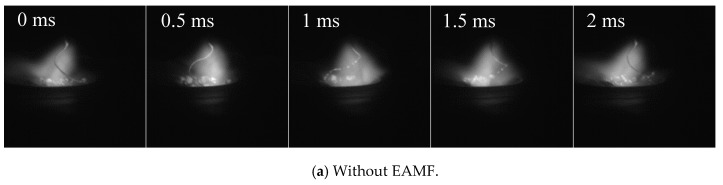
High-speed video observation of high-current GMAW metal transfer processes with EAMFs.

**Table 1 materials-13-05792-t001:** Boundary conditions.

Boundary	*v*/(m·s^−1^)	*P*/Pa	*T*/K	*Φ*/V	*A*/(Wb·m^−1^)
Gas inlet	*v* _gin_	—	300	$\frac{\partial \Phi }{\partial n}=0$	$\frac{\partial A_{i}}{\partial n}=0$
Wire inlet	*v* _min_	—	1500	$-\sigma \frac{\partial \Phi }{\partial n}=j$	$\frac{\partial A_{i}}{\partial n}=0$
Gas outlet	—	*P* _ATM_	$\frac{\partial T}{\partial n}=0$	$\frac{\partial \Phi }{\partial n}=0$	A=0
Wire wall	0	—	coupled	coupled	Coupled
Ground wall	0	—	3000	0	$\frac{\partial A_{i}}{\partial n}=0$

**Table 2 materials-13-05792-t002:** Welding parameters.

Nomenclature	Value
Current	DC 400 A
Wire diameter	1.2 mm
Wire feed rate	0.3 m/s
Arc length	10 mm
Argon shielding gas flow rate	20 L/min
Magnetic flux density	0.02 Tesla

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
