# Peer review of "3D Numerical Study of External Axial Magnetic Field-Controlled High-Current GMAW Metal Transfer Behavior"

_materials, 2020, doi:10.3390/ma13245792_

Round 1

Reviewer 1 Report

The authors report on a modelling of a GMAW process with axial magnetic field control. They developed a very powerful MHD model, which can treat the complex material transfer coupled with the arc behaviour. This includes important aspects of the arc physics as the effect of metal evaporation. Furthermore, a sophisticated model of the molten wire and droplet detachment is considered. Although, the model was already presented elsewhere recently, the application of such a detailed approach to a 3D problem including an alternating external magnetic field is a unique achievement. Furthermore, the model is able to predict the very complex movement of the molten material and the arc in the specific situation of high current and alternating axial magnetic field in sufficient agreement with experimental observations. Hence, the model allows important insides into the underlying physical mechanisms, which would be never possible with experimental methods. I congratulate the authors for this very excellent results.

The paper is well written with a detailed literature review on the topic of magnetic field control of GMAW and its modelling. The paper fits into the scope of the journal by considering a modern and effective arc welding method. There are a number of simplifications in the model, which are meaningful and probably required to limit the computational effort. I have only minor remarks for an optional improvement of the explanation of the method and the presentation of the results:

  1. The authors mention the simplification that the description of the weld pool is avoided and the surface temperature is set to 3000 K at the cathode. However, the reasons may be not well known for some of the readers, and the authors should shortly explain the simplification of the treatment of the arc cathode attachment – only the artificial increase of the conductivity above the cathode surface is mentioned so far.
  2. The treatment of the melting border of the wire electrode becomes not completely clear for me. Is this border fixed or can it change depending on the temperature distribution in the wire together with the arc anode attachment? The corresponding conditions may be significant for the description of the decrease of the arc length observed in the experiments when a magnetic field with low frequency is applied.
  3. The authors discuss the impact of the frequency of the magnetic field on the flow velocity in the arc by means of figure 11. It can be seen very well in figure 11e) that the velocity is significantly low at 500 Hz. However, this effect should be also considered in the discussion.
  4. As an option, the authors should think about possibilities to publish videos corresponding to the figures (e.g. 9 and 11), because it is hard for the reader to estimate the rotation of the material transfer as well as of the arc from the single pictures. Another option could be to show e.g. the trajectory of a center of mass in a selected cross section to make the rotation and angle deflection more clear.
  5. There are some uncommon phrases in the text, which should be checked e.g. by a native speaker or during the edition process (see e.g. row 50 - should be “is of low cost”, row 273 – should be “was intensified”, row 329 – better “at selected moments”), there is a typo error: row 202 - should be “diameter of 1.2 mm”.

Author Response

-Referee 1

The authors report on a modelling of a GMAW process with axial magnetic field control. They developed a very powerful MHD model, which can treat the complex material transfer coupled with the arc behaviour. This includes important aspects of the arc physics as the effect of metal evaporation. Furthermore, a sophisticated model of the molten wire and droplet detachment is considered. Although, the model was already presented elsewhere recently, the application of such a detailed approach to a 3D problem including an alternating external magnetic field is a unique achievement. Furthermore, the model is able to predict the very complex movement of the molten material and the arc in the specific situation of high current and alternating axial magnetic field in sufficient agreement with experimental observations. Hence, the model allows important insides into the underlying physical mechanisms, which would be never possible with experimental methods. I congratulate the authors for this very excellent results.

The paper is well written with a detailed literature review on the topic of magnetic field control of GMAW and its modelling. The paper fits into the scope of the journal by considering a modern and effective arc welding method. There are a number of simplifications in the model, which are meaningful and probably required to limit the computational effort. I have only minor remarks for an optional improvement of the explanation of the method and the presentation of the results:

  1. The authors mention the simplification that the description of the weld pool is avoided and the surface temperature is set to 3000 K at the cathode. However, the reasons may be not well known for some of the readers, and the authors should shortly explain the simplification of the treatment of the arc cathode attachment – only the artificial increase of the conductivity above the cathode surface is mentioned so far.

Respond: Thank you for your comments. We did not consider the weld pool, instead, the cathode surface temperature is set to 3000K for assuring the current passes through the cathode. The cathode effect was not considered in this model. It is a simplified treatment base on Local thermodynamic equilibrium and appropriate mesh sizes to make the current continuous in the whole domain. We have modified our manuscript in line-146.

  1. The treatment of the melting border of the wire electrode becomes not completely clear for me. Is this border fixed or can it change depending on the temperature distribution in the wire together with the arc anode attachment? The corresponding conditions may be significant for the description of the decrease of the arc length observed in the experiments when a magnetic field with low frequency is applied.

Respond: Thank you for your comments. Only 2mm length of the feeding wire is considered in the calculated domain since the calculated amount is huge if 15mm length of the feeding wire is considered. The top cross-section surface temperature is assumed to be 1500K which is a reasonable value a little less than the melting point of the wire. And the temperature of the 2mm wire electrode is depended on the arc anode attachment and its own Joule heat.

  1. The authors discuss the impact of the frequency of the magnetic field on the flow velocity in the arc by means of figure 11. It can be seen very well in figure 11e) that the velocity is significantly low at 500 Hz. However, this effect should be also considered in the discussion.

Respond: Thank you for your comments. The velocity is lower once applied a relatively higher frequency magnetic field since the alternating electromagnetic forces. We have modified our discussion about your suggested comments.

  1. As an option, the authors should think about possibilities to publish videos corresponding to the figures (e.g. 9 and 11), because it is hard for the reader to estimate the rotation of the material transfer as well as of the arc from the single pictures. Another option could be to show e.g. the trajectory of a center of mass in a selected cross section to make the rotation and angle deflection more clear.

Respond: Thank you for your comments. The videos can be provided if necessary.

  1. There are some uncommon phrases in the text, which should be checked e.g. by a native speaker or during the edition process (see e.g. row 50 - should be “is of low cost”, row 273 – should be “was intensified”, row 329 – better “at selected moments”), there is a typo error: row 202 - should be “diameter of 1.2 mm”.

    Respond: Thank you for your comments. We have revised all the pointed phrases.

Reviewer 2 Report

The manuscript shows interesting work in the field of high productive gas metal arc welding. However, the introduction does not refer to important information concerning external magnetic fields (e.g. industrially applied frequency, strength, polarity, etc.) and needs to be improved. Furthermore, the experimental setup and polarity of the described EAMF is of high interest and needs to be shown. Material data for simulation and experimental trials is missing. An examination of the total material constriction depending on the variables of the EAMF needs to be carried out and shown in a final figure.

Reviewers comments:

Line 40.   "..few related studies in the literature." References missing

Line 51. Reference missing for the long term use of EMF

Line 52. melted --> molten

Line 64.   "..GMA.." --> GMAW

Line 154. Industrially applied EMF frequencies have values < 50 Hz due to mass inertia of the transferred droplets; the analyzed frequencies seem to be very high. A reference is needed.

Line 155. Why was B set to 0.02 T? Magnetic deflection of arc and transferred droplets can be measured much earlier. A reference is needed.

Line 160. "iron specie" Wrong term. The meaning is unclear

Line 164. Please add mesh type and element size

Line 202. Wrong symbol for diameter.

Line 202. Please specify the mild steel and Ar used. Is the material data used in the simulation similar to the chemical composition of the welding wire in the  experimental trials?

Line 226. "...metal vapor evaporated" Wrong term.

Line 247 pp. the expression "arc flow velocity" is not clear and consistent throughout the manuscript. (plasma flow, shielding gas flow, material flow, etc.?)

Line 252. In the following figure a vector with 100 m/s appears in the top right corner. The explanation is missing.

Figure 10 b). "arc velocity" or "arc flow velocity" is not consistent and unclear.

Line 334. It is unclear how the polarity of the EAMF attaches. At which polarity does the effect enhance or diminish.

Line 362. A figure of the experimental Setup and polarity is missing. How and where is the EAMF induced to the GMAW process? What is the distance between magnetic unit and welding wire, respectively arc?

Line 389. Figure 11: Missing acting magnetic force (direction, intensity, magnetizing force)

Line 389. The simulation was carried out with a constant arc length of 10mm. The figures in figure 11 show a varying arc length. How was this variation reflected in the results?

Figure 11. The single images of the material transfer are too small. Please enlarge the single pictures.

Line 416. "...regardless of the EAMF frequency"? It is shown that at higher EAMF frequencys a cooling effect appears, so that the arc temperature decreases. Please add description.

Line 423. Why is there no effect for f = 500 Hz? Based on the simulation results there should be some.

Reviewer 3 Report

Dear Authors,

I have read your paper titled "3D numerical study axial magnetic field-controlled high-current GMAW metal transfer behaviour". In my opinion the topic is interesting and paper fulfill aims and scope of the Materials journal.

I have some suggestions, which may be helpfull in improving your work.

  1. Line 18; "However, this decrease the welding quality as a result ..." It has been written as an unconditional effect, and this deterioration can only occur, although it must be admitted that the risk is high.
  2. Line 139; "The arc length between the solid wire tip and the base metal surface was 10 mm". Arc length is related to voltage. Unfortunately, the article does not specify the voltage, but it can be concluded from the declared welding current (400 A) that the voltage is around 35V. Then the arc length is statistically approximately 20 mm. Please consider the correctness of the assumption 10mm arc lengt.
  3. Line 155; "The magnetic flux density waveform of the alternating EAMF was simplified to be a square wave" - the impact of this simplification can have a strong impact on the results
  4. Line 160; "iron specie" seems to be incomprehensible
  5. Fig 10; The standard deviation of the presented values is significantly greater than the difference in the mean value of the presented results. This situation requires extensive comment.
  6. General remark. The values obtained from modeling should be presented as approximate and largely uncertain. The trends of changes and the reaction of the examined system to variable external factors have a greater cognitive value than the absolute values. Especially for the extremely stochastic case of GMAW welding.
  7. The experimental verification took place in significantly different conditions, arc voltage, wire feed rate 18 m/min (20 m/min), wire stick-out length 10 mm (20 mm). Such large differences do not allow for fair comparison and verification of the modeling results. At least requires extensive commentary.
  8. General remark. A very valuable and good unified arc and droplet model was created for the 3D high current GMAW to investigate the rotation and spray metal transfer behavior. It is important and worth paper. The experimental verification of the model should take place in more similar conditions (and I mean the parameters of the welding process).

Regards

Author Response

-Referee 3

I have read your paper titled "3D numerical study axial magnetic field-controlled high-current GMAW metal transfer behaviour". In my opinion the topic is interesting and paper fulfill aims and scope of the Materials journal. I have some suggestions, which may be helpfull in improving your work.

  1. Line 18; "However, this decrease the welding quality as a result ..." It has been written as an unconditional effect, and this deterioration can only occur, although it must be admitted that the risk is high.

Respond: Thank you for your kind suggestion. We use ‘much higher current’ instead of the previous statement.

  1. Line 139; "The arc length between the solid wire tip and the base metal surface was 10 mm". Arc length is related to voltage. Unfortunately, the article does not specify the voltage, but it can be concluded from the declared welding current (400 A) that the voltage is around 35V. Then the arc length is statistically approximately 20 mm. Please consider the correctness of the assumption 10mm arc length.

Respond: Thank you for your kind suggestion. In the experimental case, the GMAW power source is a constant voltage output, and the welding current depends on the wire feed rate. In the numerical treatment, we can only measure the welding current and set it as a boundary condition for calculating. It is a simplified treatment for calculated feasibility. And the arc length can only be assumed as a constant value depends on our limited modeling ability. The arc length is not 20mm, the wire stick-out length is 20mm.

3.Line 155; "The magnetic flux density waveform of the alternating EAMF was simplified to be a square wave" - the impact of this simplification can have a strong impact on the results

Respond: Thank you for your comments. It is a simplified treatment as described in our manuscript. In the future, we will pay attention to those differences.

  1. Line 160; "iron specie" seems to be incomprehensible

Respond: Thank you for your comments. It is iron vapor since only iron is considered in the metal vapor. We will modify our statement in the text.

  1. Fig 10; The standard deviation of the presented values is significantly greater than the difference in the mean value of the presented results. This situation requires extensive comment.

Respond: Thank you for your comments. This fluctuation is expressed by the standard deviations, and it was found that the highest fluctuations in arc temperature and flow velocity when the low-frequency 100 Hz magnetic field was applied. We added some comments on that in our manuscript.

General remark. The values obtained from modeling should be presented as approximate and largely uncertain. The trends of changes and the reaction of the examined system to variable external factors have a greater cognitive value than the absolute values. Especially for the extremely stochastic case of GMAW welding.

The experimental verification took place in significantly different conditions, arc voltage, wire feed rate 18 m/min (20 m/min), wire stick-out length 10 mm (20 mm). Such large differences do not allow for fair comparison and verification of the modeling results. At least requires extensive commentary.

General remark. A very valuable and good unified arc and droplet model was created for the 3D high current GMAW to investigate the rotation and spray metal transfer behavior. It is important and worth paper. The experimental verification of the model should take place in more similar conditions (and I mean the parameters of the welding process).

Respond: Thank you for your comments. The arc length is measured to be around 5-10mm, and we assume the arc length is about 10mm in the numerical model. And it is very difficult for us to make the experimental and numerical conditions to be absolutely the same since it’s a complex system. We will improve our mathematic model and experimental conditions in our next works.

Reviewer 4 Report

         The obtained results are of great interest to the readers of this valuable journal Materials. A comparison between current literature and previous studies shows that the results from this manuscript are new and correct.

I suggest to the Editor-in-Chief accepting this manuscript, but after the authors make the following corrections:

  1. Please highlight how the work advances or increments the field from the present state of knowledge and provide a clear justification for your work;
  2. Please explain more clearly the main Eqs. (1)-(5) and put point (dot) after equations (2)-(9) and at the end of line 104.
  3. I suggest being heightened more clearly the contributions of the authors in the Introduction and Conclusions.
  4. The references are adequate and all they are necessary. I think the authors must strengthen the References section with titles that use similar numerical methods and simulation techniques, as in the manuscript proposed model, for instance:
  • Numerical Simulation of Arc and Droplet Behaviors in TIG-MIG Hybrid Welding. Materials 13, 4520, (2020), DOI:10.3390/ma13204520;
  • Weaker hypotheses for the general projection algorithm with corrections St. Univ." Ovidius" Constanta-Seria Matematica 23 (3), 9-16, (2015), DOI: 10.1515/auom-2015-0043.

Summarizing, I recommend the acceptance of this paper after Minor Revisions.

Author Response

-Referee 4

The obtained results are of great interest to the readers of this valuable journal Materials. A comparison between current literature and previous studies shows that the results from this manuscript are new and correct.

I suggest to the Editor-in-Chief accepting this manuscript, but after the authors make the following corrections:

  1. Please highlight how the work advances or increments the field from the present state of knowledge and provide a clear justification for your work;

Respond: Thank you for your comments. We have highlighted its advances, especially for its high-efficiency and controllability.

  1. Please explain more clearly the main Eqs. (1)-(5) and put point (dot) after equations (2)-(9) and at the end of line 104.

Respond: Modified as suggested.

  1. I suggest being heightened more clearly the contributions of the authors in the Introduction and Conclusions.

Respond: Thank you for your comments. We modified our introduction by adding some necessary references.

4.The references are adequate and all they are necessary. I think the authors must strengthen the References section with titles that use similar numerical methods and simulation techniques, as in the manuscript proposed model, for instance:

Numerical Simulation of Arc and Droplet Behaviors in TIG-MIG Hybrid Welding. Materials 13, 4520, (2020), DOI:10.3390/ma13204520;

Weaker hypotheses for the general projection algorithm with corrections St. Univ." Ovidius" Constanta-Seria Matematica 23 (3), 9-16, (2015), DOI: 10.1515/auom-2015-0043.

Respond: Thank you for your comments. We added some necessary references for improving the manuscript.

Round 2

Reviewer 2 Report

Thank you for the improvement of the manuscript.

Please add the reference of your previous paper with a short description of the experimental setup of the axial magnetic field to chapter 3.2.

Author Response

Thanks for your comments. The experimental setup was described as modifying.

'The electromagnetic coil excited by a self-made excitation source was covered on the coaxial welding torch nozzle to generate the EAMF[18].'

Reviewer 3 Report

Dear Authors, 

Thank you very much for your kind and substantive reply to my comments. After making corrections, I recommend accepting the manuscript in present form.

Regards,

Author Response

Thank you for your comments.

We have modified our manuscript as you suggested.